# SEDS: Semantically Enhanced Dual-Stream Encoder for Sign Language Retrieval

## ABSTRACT

Sign language retrieval, as an emerging visual-language task, has received widespread attention. Different from traditional video retrieval, it is more biased towards understanding the semantic information of human actions contained in video clips. Previous works typically only encode RGB videos to obtain high-level semantic features, resulting in local action details drowned in a large amount of visual information redundancy. Furthermore, existing RGB-based sign retrieval works suffer from the huge memory cost of dense visual data embedding in end-to-end training, and adopt offline RGB encoder instead, leading to suboptimal feature representation. To address these issues, we propose a novel sign language representation framework called Semantically Enhanced Dual-Stream Encoder (SEDS), which integrates Pose and RGB modalities to represent the local and global information of sign language videos. Specifically, the Pose encoder embeds the coordinates of keypoints corresponding to human joints, effectively capturing detailed action features. For better context-aware fusion of two video modalities, we propose a Cross Gloss Attention Fusion (CGAF) module to aggregate the adjacent clip features with similar semantic information from intra-modality and inter-modality. Moreover, a Pose-RGB Fine-grained Matching Objective is developed to enhance the aggregated fusion feature by contextual matching of fine-grained dual-stream features. Besides the offline RGB encoder, the whole framework only contains learnable lightweight networks, which can be trained end-to-end. Extensive experiments demonstrate that our framework significantly outperforms state-of-the-art methods on How2Sign, PHOENIX-2014T, and CSL-Daily datasets.

## CCS CONCEPTS

• **Information systems** → **Multimedia and multimodal retrieval**.

## KEYWORDS

Sign language retrieval, Multimodal alignment, Feature fusion

**ACM Reference Format:**
Anonymous Author(s). 2024. SEDS: Semantically Enhanced Dual-Stream Encoder for Sign Language Retrieval. In *Proceedings of the 32th ACM International Conference on Multimedia (MM '24), October 28–November 1, 2024, Melbourne, Australia.* ACM, New York, NY, USA, 10 pages. https://doi.org/XXXXXXX.XXXXXXX

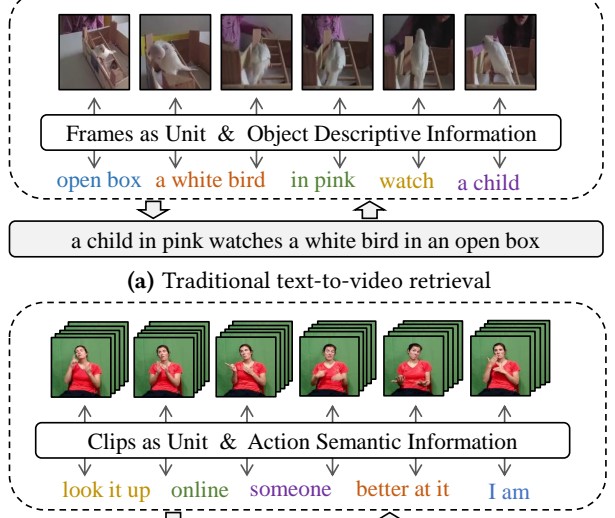

**(a)** Traditional text-to-video retrieval

**(b)** Sign language text-to-video retrieval

**Figure 1: The difference in properties of video information between (a) traditional text-to-video retrieval and (b) sign language text-to-video retrieval. The former describes the content in video frames, while the latter corresponds to the semantics of the actions in video clips.**

## 1 INTRODUCTION

Sign language is the primary means of communication for people with hearing impairments. Sign language understanding is an important research field dedicated to overcoming communication barriers between hard-of-hearing individuals and non-signers. The classic sign language comprehension task [11, 12, 18, 21, 26, 27, 48, 57–59, 64] includes sign language recognition and translation, intending to identify the gloss sequence in sign language videos and further translate it into natural language. In this paper, we focus on the emerging sign language retrieval task [10, 15]. Unlike sign language recognition and translation, sign language retrieval task is targeted at retrieving videos or texts that are most similar to the query from a video or text database.

As a special case of traditional text-video retrieval (TVR) tasks [14, 17, 19, 30, 39–41, 43, 55, 63], sign language retrieval comprises two sub-tasks, *i.e.*, text-to-sign-video (T2V) retrieval and sign-video-to-text (V2T) retrieval. Unlike traditional TVR tasks, sign TVR focuses on the semantic information of human actions that correspond to natural language words contained in video clips (Fig. 1(b)), rather than descriptive information of objects in video frames (Fig. 1(a)). Sign language retrieval task is more similar to translating a language

expressed through video clips, and then conducting cross-language retrieval [10]. Therefore, in sign language videos, a video clip containing a single action is the smallest unit of complete information, which can be defined as a sign visual word. This requires the presence of a sign encoder in the model to extract action semantic information from video clips.

Recently, as a common practice for text-to-video retrieval models, contrastive loss [22] is involved in training, which significantly improves performance by utilizing visual prior knowledge from the pre-trained CLIP vision transformer [51]. In contrastive learning, multiple negative samples are required in a single batch. However, previous sign language retrieval works, such as SPOT-ALIGN [15] and CiCo [10], use RGB-based sign encoders that embed dense visual data. The massive memory cost results in two-stage training instead of end-to-end training, which potentially reduces the quality of the feature representation. Meanwhile, sign language usually utilizes two types of signals, *i.e.,* global visual signals including body position and facial expressions, and local action signals including hand actions and palm movements. Therefore, the sign encoder needs to pay attention to both the semantics of the global visual information and the local action information. In contrast, RGB encoders always embed global visual signals to obtain high-level semantic features, leading to local action details drowned in a large amount of visual information redundancy. Apart from the lack of local information, such bias also results in potential robustness issues, affecting the performance of model due to the difference in video backgrounds or signers between the training and test sets.

In order to alleviate the above issues, inspired by previous sign language tasks [18, 24, 29, 35], we introduce Pose modal knowledge to sign language retrieval task and propose a new framework called Semantically Enhanced Dual-Stream Encoder (SEDS), which includes an online Pose encoder and an offline RGB encoder extracting features from two different perspectives. Compared to the offline RGB encoder, the remaining modules within the framework such as Pose encoder, fusion module and interaction transformers, are lightweight enough to be trained end-to-end, which effectively improves the feature quality of sign language videos. The introduction of Pose keypoints of both hand parts and entire body skeleton not only enables the Pose encoder to supplement the details of local action movement, but also effectively mitigates the bias for similar visual scenes introduced by the RGB modality. For the process of fusing RGB and Pose modalities, inspired by recent work GASLT [60], we propose the Cross Gloss Attention Fusion (CGAF) module. This module keeps its attention on video clips locally, aggregating adjacent clips with similar semantic information from intra-modal and inter-modal. This enhances the semantic representation of sign visual words in video clips.

To ensure optimal retrieval performance through fusion features, we propose an explicitly supervised Pose-RGB fine-grained matching objective. This objective performs contextual matching on fine-grained dual-stream features, matching the corresponding clip features in the two modalities. Therefore, it ensures that the same clips in both modalities have a higher chance of matching the same word feature with more similar semantics. The distribution of the fine-grained similarity matrices of Pose-Text and RGB-Text are implicitly aligned, allowing the CGAF module to fully consider the representations from both modalities during the fusion process.

The main contributions of our work are summarized as follows:

- To address the limitation of previous sign language retrieval works, we propose a framework called Semantically Enhanced Dual-Stream Encoder (SEDS) that first introduces Pose modality knowledge into sign language retrieval task.
- A Cross Gloss Attention Fusion (CGAF) module is applied to the fusion of Pose and RGB modalities, which fuses local information from intra-modal and inter-modal to enhance the representation ability of fusion clip features.
- To further improve the representational ability of the fusion features, a supervised Pose-RGB fine-grained matching objective is designed for training to implicitly align the fine-grained similarity matrix of Pose-Text and RGB-Text.
- The extensive experiments and evaluations demonstrate that our framework significantly outperforms state-of-the-art methods on How2Sign, PHOENIX-2014T, and CSL-Daily sign language retrieval datasets.

## 2 RELATED WORK

### 2.1 Sign Language Understanding

Sign language understanding aims to fully understand the semantic information in sign videos and connect it with natural language. The current mainstream sign language understanding tasks include isolated sign language recognition [25, 33–35, 61, 68], continuous sign language recognition [11, 12, 21, 26, 27, 48], sign language translation [18, 57–59, 64, 66], sign language spotting [1, 20, 46], and our focused sign language retrieval [10, 15] task.

As the foundation of sign language understanding, sign language recognition task aims to convert an input sign language video into a gloss sequence, which corresponds to the sign visual words in natural language. The encoder used for extracting sign language video features is mainly based on the CNN [32] architecture, including 3D-CNN [8, 36, 49] and 2D+1D CNN [13, 66]. The sign language spotting task [1, 20, 46] is a variant of the sign language recognition task, aimed at locating and recognizing sign word instances in an untrimmed video. The spotting features can be used as features of sign visual words, matching with the sign language annotation text at a fine-grained level. Therefore, we leverage the I3D [6] network pre-trained on BSL-1K [20] for sign spotting as offline RGB encoder.

For sign language understanding tasks, there are two types of methods: gloss-based [5, 8, 9, 53, 65, 66] and gloss-free [2, 36, 44, 60, 62] methods. The difference between them is whether or not to use gloss annotations as auxiliary supervision. Recently, GASLT [60] argues that gloss plays a role in restricting attention to local sequences in sign language tasks, which is more consistent with logicality of sign language videos. Inspired by it, we propose Cross Gloss Attention Fusion (CGAF) module, which aggregates local information with similar semantics from intra-modal and inter-modal.

### 2.2 The Role of Pose in Sign Language Tasks

Pose is an important modal information in the process of sign language video understanding, which is generally extracted offline by the Pose extraction model [28, 52, 56]. Compared to RGB data, Pose has less redundant information and is more targeted towards the position and motion information of human movements. There are many methods using Pose information. Works like [47, 66] crop

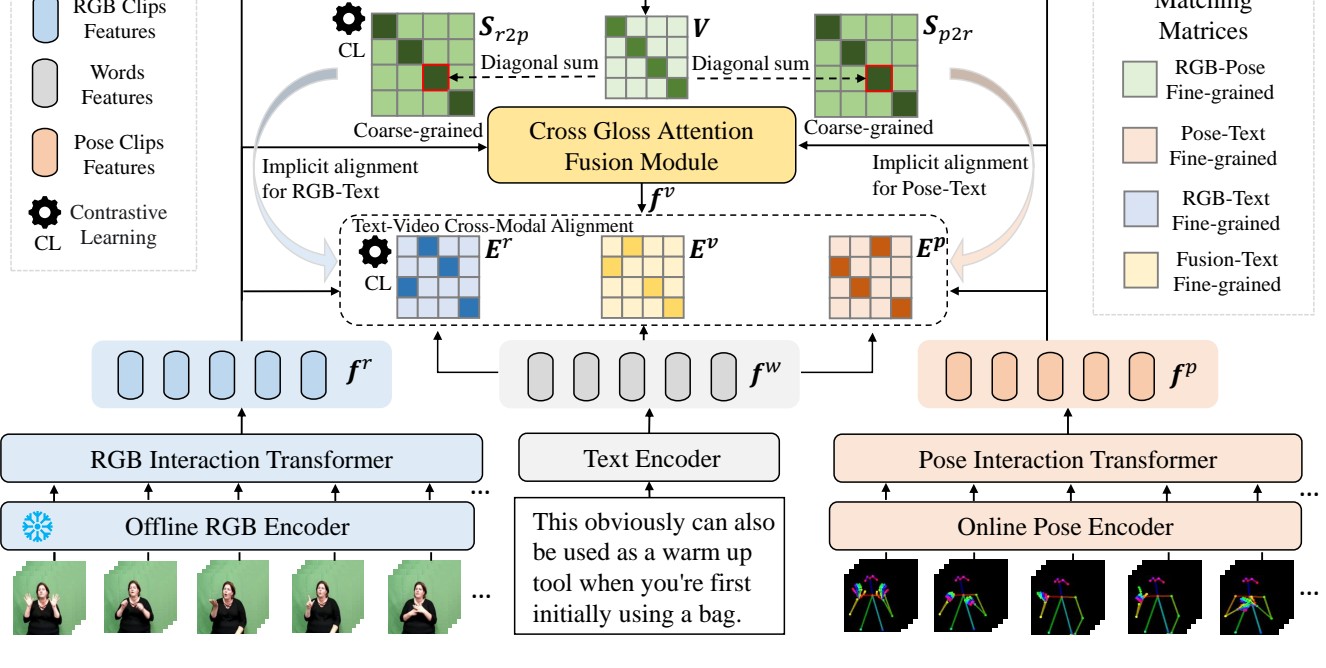

**Figure 2: The overview of our Semantically Enhanced Dual-Stream Encoder framework, which consists of three parts: 1) Pose and RGB Feature Extraction Module. 2) Cross Gloss Attention Fusion (CGAF) Module. 3) Pose-RGB Fine-grained Matching Objective. The network is jointly optimized by: 1) three text-video alignment losses between Pose, RGB, Fusion features and Text features. 2) an RGB-Pose fine-grained alignment loss between Pose modality and RGB modality.**

RGB videos and feature maps into different parts based on Pose keypoints. Some other works [9, 67] use Pose heatmap to supplement the human skeleton information in RGB, while sharing the architecture with the RGB encoder. However, due to their strong correlation with RGB, these methods still cost excessive memory. Following previous work [4, 24, 29], we use Graph Convolutional Networks (GCN) to model Pose keypoints into a graph, defining edges based on the connections between human skeleton joints. This allows us to extract features that represent precise local position and motion information of the human body. Furthermore, the sparse input of Pose keypoints enables end-to-end training.

## 2.3 Sign Language Text-video Retrieval

Traditional text-video retrieval [14, 17, 19, 30, 39–41, 55, 63] is a fundamental vision-language task, which primarily focuses on videos of more general daily life scenarios. Early works [14, 17, 39, 40, 55] mainly align the features of text and video by training text and video encoders from scratch. Due to the great success of the large-scale image-text pre-training model CLIP [50], many CLIP-based text-video retrieval works [19, 30, 41, 63] have achieved significant performance improvements.

Sign language text-video retrieval can be seen as a special case of text-video retrieval. CLIP-based text-video retrieval [43] mainly involves sparse frame sampling of the original video, but sign language videos often require multiple frames to express a sign visual word. Therefore, before sending it to the vision transformer, a sign encoder is needed to extract clip features. Lots of CLIP-based

works [37, 38, 45, 54] have proposed multi-grained modal alignment methods, including both supervised coarse-grained and fine-grained feature alignment, fine-grained aggregation feature alignment, etc. However, our proposed Pose-RGB fine-grained matching objective is different from these methods. It matches the contextual features of two video modalities at a fine-grained level, implicitly aligning the distribution of fine-grained similarity matrices between Pose, RGB and Text. This allows for a full consideration of the characteristics of both modalities during the fusion process.

## 3 METHOD

In this section, we introduce our Semantically Enhanced Dual-Stream Encoder (SEDS) framework (Fig. 2), which is composed of three parts: 1) Pose and RGB Feature Extraction Module, which extracts local action information and global visual information. 2) Cross Gloss Attention Fusion (CGAF) Module, which fuses the adjacent clip information from both inter-modal and intra-modal perspectives. 3) Pose-RGB Fine-grained Matching Objective, ensuring the contextual alignment of fine-grained dual-stream features.

## 3.1 Pose and RGB Feature Extraction Module

**Sampling Clips in Sign Video.** Sign language videos and natural language serve as two distinct modalities of expression with different word orders, yet they convey the same semantics. The fundamental sign visual words depicted in sign language videos should correspond with words in natural language. For a single video, we begin by filtering out low-quality video frames, such as

those with unclear or missing hand gestures and incomplete signers. Subsequently, we utilize a sliding window approach with a window size of 16 and a sliding interval of 1 to generate a number of video clips, and then select $T$ clips at equal intervals in time dimension. Each video clip is then fed into online Pose and offline RGB sign encoders, resulting in Pose features $f^{p\prime} \in \mathbb{R}^{T \times D}$ and RGB features $f^{r\prime} \in \mathbb{R}^{T \times D}$ for this sign video. Next, we feed these two sequence features separately into two Transformer encoders for interaction in their respective modalities, finally obtaining features $f^p \in \mathbb{R}^{T \times D}$ and $f^r \in \mathbb{R}^{T \times D}$ for the further fusion.

**Online Pose Encoder.** In order to obtain the Pose data in the video, we use the open-source RTMPose [28] to extract the Pose data from the original RGB sign language video. To better utilize the motion information of different skeletal parts of the human body, we divide the extracted Pose keypoints into three groups: 21 keypoints $G_L$ for left hand, 21 keypoints $G_R$ for right hand, and 7 keypoints $G_B$ for body. We use two GCN network modules for feature extraction of Pose data and one 1D convolutional fusion module for information fusion in the time dimension in Pose encoder. We share the same GCN network for both hands. In the GCN module, we use edges to represent the connections of the human skeleton in hand and body group, further constructing the group-specific adjacency matrix $A_G$. The single-layer convolution operation for a specific group of keypoint features is as follows:

$$f_s^L = \sigma \left( \Lambda_G^{-\frac{1}{2}} \left( A_G + I \right) \Lambda_G^{-\frac{1}{2}} f_s^{L-1} W_G^{L-1} \right), \quad (1)$$

where $f_s^L$ denotes the $L$-layer feature vector of the corresponding $s$ frame, and the feature vector is the original coordinates of keypoints when $L = 0$. $W_G^{L-1}$ is the $L-1$-layer weight matrix for performing feature linear transformation, and $\sigma$ is a nonlinear transformation. $I$ represents the self-connection of points. $\Lambda_G^{ii}$ is the normalized diagonal matrix $\sum_j (A_G^{ij} + I^{ij})$.

After extracting spatial features from all frames of a single video, we obtain three groups of features $f_L^s$, $f_R^s$, $f_B^s \in \mathbb{R}^{F \times D}$, where $F$ represents the number of frames. Then we concatenate them into feature $f^s \in \mathbb{R}^{F \times 3D}$. Next, based on the pre-defined video clip boundaries, we sample these frame features into many sets of clip features $f^{s\prime} \in \mathbb{R}^{T \times 16 \times 3D}$ containing 16 frame features. In the temporal dimension, we use a 1D convolutional fusion module to aggregate information within clips. After information aggregation, the features $f^{p\prime} \in \mathbb{R}^{T \times D}$ will be ultimately fed to a Transformer encoder for intra-modal information interaction.

**Offline RGB Encoder.** Meanwhile, there have been significant advances in pre-trained sign spotting [20, 46] research. When applied to downstream tasks, these RGB-based models [20] greatly enhance the understanding capabilities of convolutional neural networks for sign language videos. Therefore, we employ an I3D network pre-trained on BSL-1K [1] as the offline RGB encoder to extract $f^{r\prime} \in \mathbb{R}^{T \times D}$ features. If the RGB encoder performs end-to-end learning simultaneously with the Pose encoder, the semantics of specific sign language video can be better captured. However, in the actual training process, extracting features from such contrastive video samples consumes a considerable amount of memory. Consequently, we freeze the pre-trained I3D network and extract features from each sign language video clip offline.

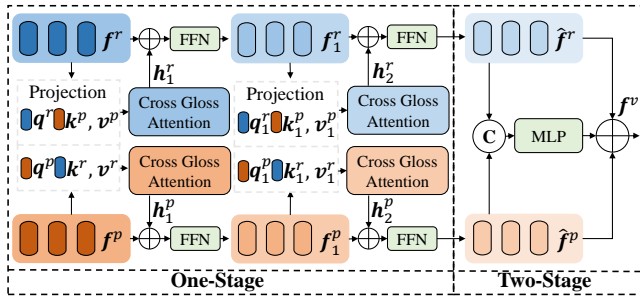

**Figure 3: The structure of two-stream Cross Gloss Attention Fusion (CGAF) module.**

## 3.2 Cross Gloss Attention Fusion Module

For the process of fusing Pose and RGB modalities, there are some naive fusion methods such as concatenation or cross-attention. However, those naive methods ignore the logic and coherence of sign language videos. Specifically, the semantic relevance of adjacent clips in sign language videos is significantly higher than distant clips. Inspired by a new attention mechanism called gloss attention [60] in recent sign language translation tasks, we propose a Cross Gloss Attention Fusion (CGAF) module. Compared to simple addition or concatenation, our module emphasizes the attention interaction between local information rather than global information including irrelevant distant features, allowing for context-aware integration of features from inter-modal and inter-modal.

The CGAF Module combines the Pose modality and RGB modality features of sign language videos in two main stages. The first stage aggregates local information between the two modalities, while the second stage merges the features obtained after aggregation. In the first stage, we start by obtaining the Pose vectors $q^p, k^p, v^p \in \mathbb{R}^{T \times D}$ and RGB vectors $q^r, k^r, v^r \in \mathbb{R}^{T \times D}$, which are computed as linear projections of the Pose modality feature $f^p$ and RGB modality features $f^r$. Similar to traditional cross attention modules, we divide these two groups of vectors into two groups $\{q^p, k^r, v^r\}$ and $\{q^r, k^p, v^p\}$. Taking group $\{q^p, k^r, v^r\}$ as an example, we first obtain $T \times N$ constant attention positions $P^p \in \mathbb{R}^{T \times N}$ for each query in $q^p$. Then we compute $N$ dynamical offset through $O^p = W_o^p q^p \in \mathbb{R}^{T \times N}$, where $W_o^p \in \mathbb{R}^{D \times N}$. Due to the adjusted attention position $\hat{P}^p = (P^p + O^p)\%T$ is a floating-point number, we use linear interpolation sampling on $k^r, v^r \in \mathbb{R}^{T \times D}$ to obtain the new $\hat{k}^r, \hat{v}^r \in \mathbb{R}^{T \times N \times D}$. The final calculation equation is as follows:

$$h_t^p = \sum_{i=1}^{N} \hat{v}_{t,i}^r \cdot \frac{\exp(q_t^p \cdot \hat{k}_{t,i}^r)}{\sum_{j=1}^{N} \exp(q_t^p \cdot \hat{k}_{t,j}^r)}, \quad (2)$$

where $h^p \in \mathbb{R}^{T \times D}$ is the attention vector obtained after cross gloss attention calculation. The method for calculating attention vectors for group $\{q^r, k^p, v^p\}$ is also the same. In the first stage, we use two layers of cross gloss attention to obtain the final output features $\hat{f}^p, \hat{f}^r \in \mathbb{R}^{T \times D}$, as shown in Fig. 3.

In the second stage, the dual stream features $\hat{f}^p$ and $\hat{f}^r$ are concatenated and fed into a Multi-layer Perceptron (MLP) for interaction. Then these three features are added together to obtain the

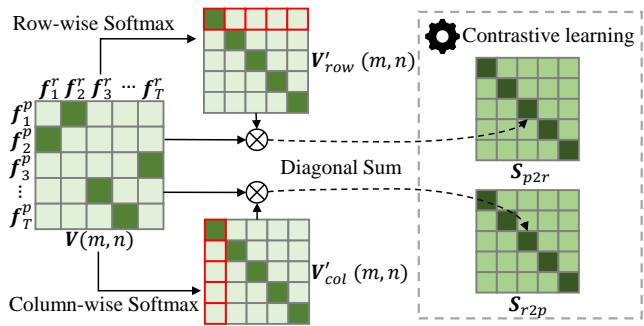

**Figure 4: The illustration of Pose-RGB fine-grained matching objective. The red box represents the softmax operation on the corresponding element.**

final feature $f^v$. The entire operation is as follows:

$$f^v = MLP([\hat{f}^p, \hat{f}^r]) + \hat{f}^p + \hat{f}^r, \tag{3}$$

where $[\cdot, \cdot]$ represents the concatenation operation, $f^v$ is the feature obtained after fusion.

### 3.3 The Learning Process of Our Framework

**Text-Video Cross-Modal Alignment.** For each sign language video, we have obtained three sets of features: Pose features $f^p$, RGB features $f^r$, and Fusion features $f^v$ from the CGAF Module. For annotation of length $L$, we use a Text encoder initialized with CLIP for feature extraction, resulting in word features $f^w \in \mathbb{R}^{L \times D}$.

According to the CiCo [10] approach for text-video cross-modal alignment, we start the matching process by choosing clip features $f_i^k$ and word features $f_j^w$, and computing a fine-grained similarity matrix $E^k(i, j) = f_i^k \cdot [f_j^w]^T \in \mathbb{R}^{T \times L}$, where $k \in \{p, r, v\}$ indicating three types of video features. Subsequently, we normalize the columns and rows of $E^k(i, j)$ using the softmax function to obtain $E_{col}^k(i, j)$, $E_{row}^k(i, j)$, and derive $E_{t2k}(i, j) = E^k(i, j) \cdot E_{col}^k(i, j)$ and $E_{k2t}(i, j) = E^k(i, j) \cdot E_{row}^k(i, j)$. Finally, summing the elements across columns and rows yields $E_{t2k}'(i, j) \in \mathbb{R}^L$ and $E_{k2t}'(i, j) \in \mathbb{R}^T$, whose averages are used as the similarity scores between text and video, $M_{t2k}(i, j)$ and $M_{k2t}(i, j)$.

For a set of $B$ text-video pairs, this method efficiently generates a pair of matrices that capture the text-video and video-text similarities, denoted as $M_{t2k}, M_{k2t} \in \mathbb{R}^{B \times B}$. We employ the InfoNCE loss [23] as text-video cross-modal alignment training objective, which is as follows:

$$\mathcal{L}_{t2k} = -\frac{1}{B} \sum_{i=1}^{B} \log \frac{\exp\left(\tau \cdot M_{t2k}^{ii}\right)}{\sum_{j=1}^{B} \exp\left(\tau \cdot M_{t2k}^{ij}\right)},$$

$$\mathcal{L}_{k2t} = -\frac{1}{B} \sum_{i=1}^{B} \log \frac{\exp\left(\tau \cdot M_{k2t}^{ii}\right)}{\sum_{j=1}^{B} \exp\left(\tau \cdot M_{k2t}^{ji}\right)},$$

$$\mathcal{L}_{t-k} = \frac{1}{2} \left(\mathcal{L}_{t2k} + \mathcal{L}_{k2t}\right),$$

where $\tau$ is a learnable temperature parameter. For the three groups of the similarity matrices, we calculate the InfoNCE loss function

separately and then sum them:

$$\mathcal{L}_{tva} = \mathcal{L}_{t-v} + \alpha \left(\mathcal{L}_{t-p} + \mathcal{L}_{t-r}\right), \tag{4}$$

where $\alpha$ is used to balance the proportion of auxiliary loss in text-video alignment. During the actual inference process, we only use the fusion features to match the text.

**Pose-RGB Fine-grained Matching Objective.** The Pose modality branch and RGB modality branch are essentially learned independently. This leads to the significant difference between the distributions of Pose-Text and RGB-Text fine-grained matching matrices, which results in poor fusion performance due to the same clips between modalities may match with different word features. To address the issue, explicit supervision on fine-grained matching is required. However, the absence of clear labels in fine-grained clip-word matching is a problem. Therefore, we utilize the two video modality features for auxiliary supervision. The video clips are represented by Pose and RGB features from two different encoders, allowing for fine-grained video matching with explicit supervision at clip-clip level, which implicitly aligns Pose and RGB modalities with text at clip-word level.

Specifically, we use Pose clip feature $f_m^p$ and RGB clip feature $f_n^r$ to calculate a fine-grained similarity matrix $V(m, n) \in \mathbb{R}^{T \times T}$. Then we perform softmax on the columns and rows of $V(m, n)$ separately to obtain $V_{col}'(m, n)$, $V_{row}'(m, n)$, and calculate two new matrices, $V_{p2r}(m, n) = V(m, n) \cdot V_{col}'(m, n)$ and $V_{r2p}(m, n) = V(m, n) \cdot V_{row}'(m, n)$, weighting the similarity score of clips. Afterwards, we sum up similarity scores on diagonal of these two matrices to obtain Pose-RGB similarity at video-video level. The process is as follows:

$$S_{p2r}(m, n) = \sum_{i=1}^{T} diag \left[V_{p2r}^{ii}(m, n) \cdot \frac{\exp\left(V_{p2r}^{ii}(m, n)\right)}{\sum_{j=1}^{T} \exp\left(V_{p2r}^{ij}(m, n)\right)}\right], \tag{5}$$

$$S_{r2p}(m, n) = \sum_{i=1}^{T} diag \left[V_{r2p}^{ii}(m, n) \cdot \frac{\exp\left(V_{r2p}^{ii}(m, n)\right)}{\sum_{j=1}^{T} \exp\left(V_{r2p}^{ji}(m, n)\right)}\right], \tag{6}$$

where $S_{p2r}$ is the Pose-to-RGB similarity matrix, and $S_{r2p}$ is the RGB-to-Pose similarity matrix. $diag[\cdot]$ represents taking only diagonal elements of a matrix.

The summing of the diagonals in the fine-grained similarity matrix is conducted because the elements on the diagonals represent the similarity between different modalities for the corresponding clips. So this process can be considered as supervised learning with clip-level labels. As a result, the corresponding clips in the Pose and RGB modalities are matched, and their features are aligned. This increases the likelihood of the same clips in both modalities matching the same word features, implicitly aligning the distributions of the fine-grained similarity matrices of Pose-Text and RGB-Text.

Upon acquiring video-video level similarity, the InfoNCE [23] loss is employed. Like the loss of text-video alignment, the loss of video-video alignment $\mathcal{L}_{p-r}$ consists of two parts, $\mathcal{L}_{p2r}$ and $\mathcal{L}_{r2p}$. These two parts correspond to $S_{p2r}$ and $S_{r2p}$. The whole process of Pose-RGB fine-grained matching objective is shown in Fig. 4.

**Joint Loss Optimization for SEDS.** We have established the text-video cross-modal alignment objective and Pose-RGB fine-grained matching objective, the overall loss function during training

| Model | T2V | | | | V2T | | | |
|---|---|---|---|---|---|---|---|---|
| | R@1↑ | R@5↑ | R@10↑ | MedR↓ | R@1↑ | R@5↑ | R@10↑ | MedR↓ |
| SA-SR [15] | 18.9 | 32.1 | 36.5 | 62.0 | 11.6 | 27.4 | 32.5 | 69.0 |
| SA-CM [15] | 24.3 | 40.7 | 46.5 | 16.0 | 17.9 | 40.1 | 46.9 | 14.0 |
| SA-COMB [15] | 34.2 | 48.0 | 52.6 | 8.0 | 23.6 | 47.0 | 53.0 | 7.5 |
| CiCo [10] | 56.6 | 69.9 | 74.7 | 1.0 | 51.6 | 64.8 | 70.1 | 1.0 |
| **Ours** | **62.5** | **75.1** | **80.1** | **1.0** | **57.9** | **70.4** | **74.9** | **1.0** |

Table 1: Different methods on How2Sign [16] dataset.

of our framework is as follows:

$$\mathcal{L} = \mathcal{L}_{tva} + \beta\mathcal{L}_{p-r}, \quad (7)$$

where $\beta$ is the proportion of Pose-RGB fine-grained alignment loss.

## 4 EXPERIMENTS

### 4.1 Setups

**Datasets.** Based on the dataset settings of previous sign language video retrieval work [10, 15], we evaluate our model on the How2Sign [16], CSL-Daily [65], and PHOENIX-2014T [3] datasets. How2Sign is a large-scale continuous American Sign Language (ASL) dataset. After removing invalid text-video pairs, we retain 31019, 1738, and 2348 available pairs in the training, validation, and testing sets. CSL-Daily is a Chinese sign language (CSL) dataset that mainly focuses on people's daily lives. It includes 18401, 1077, and 1176 available examples in the training, validation, and testing sets. PHOENIX-2014T is a German sign language (DGS) dataset that mainly includes weather forecast content from TV programs. It consists of 7096, 519, and 642 video text pairs in training, validation, and testing sets.

**Evaluation Metrics.** Following prior works [10, 15, 19, 41, 43], we use standard text-video retrieval metrics to measure retrieval performance, specifically Recall at K (R@K, higher is better) and Median Rank (MedR, lower is better). R@K denotes the proportion of correct results retrieved among the top K videos. We use K=1,5,10 in actual evaluation. MedR represents the median ranking of the correct options for all queries.

**Implementation Details.** The offline RGB encoder is an I3D [7] network pre-trained on BSL-1K [20]. The online Pose encoder consists of two GCN modules and a temporal convolution module. The GCN module of both hands is initialized using Signbert [24]. For the RGB Interaction Transformer and Pose Interaction Transformer, we set the number of layers to 12 with a hidden size of 768 and initialize it with the image encoder of CLIP (ViT-B/32) [50], but the input 1D convolution kernel is changed to the 1024 and 1536 channels corresponding to the RGB encoder and Pose encoder, then the feature channel is reduced to the same 512 channel as the vision token. For the Text encoder, we use the CLIP (ViT-B/32) text encoder for initialization. For all the datasets, we change the frame rate of sign language videos to 24 FPS, sample a maximum of 64 clips per video, and set the maximum length of words to 32. We use an Adam [31] optimizer with a cosine warm-up strategy [42]. During the training process, we freeze the offline RGB encoder, set the learning rates of the online Pose encoder and Cross Gloss Attention Fusion module to 1e-4, and set the learning rates of two Interaction Transformers and Text encoder to 1e-5. We set the batch size to 128 and train for 200 epochs. The hyper-parameters $\alpha$ in Eq. 4 is 0.8 and the $\beta$ in Eq. 7 is 0.4.

| Model | T2V | | | | V2T | | | |
|---|---|---|---|---|---|---|---|---|
| | R@1↑ | R@5↑ | R@10↑ | MedR↓ | R@1↑ | R@5↑ | R@10↑ | MedR↓ |
| SA-SR [15] | 30.2 | 53.1 | 63.4 | 4.5 | 28.8 | 52.0 | 60.8 | 56.1 |
| SA-CM [15] | 48.6 | 76.5 | 84.6 | 2.0 | 50.3 | 78.4 | 84.4 | 1.0 |
| SA-COMB [15] | 55.8 | 79.6 | 87.2 | 1.0 | 53.1 | 79.4 | 86.1 | 1.0 |
| CiCo [10] | 69.5 | 86.6 | 92.1 | 1.0 | 70.2 | 88.0 | 92.8 | 1.0 |
| **Ours** | **76.8** | **91.7** | **95.3** | **1.0** | **78.7** | **92.5** | **95.2** | **1.0** |

Table 2: Different methods on PHOENIX-2014T [3] dataset.

| Model | T2V | | | | V2T | | | |
|---|---|---|---|---|---|---|---|---|
| | R@1↑ | R@5↑ | R@10↑ | MedR↓ | R@1↑ | R@5↑ | R@10↑ | MedR↓ |
| CiCo [10] | 75.3 | 88.2 | 91.9 | 1.0 | 74.7 | 89.4 | 92.2 | 1.0 |
| **Ours** | **85.8** | **94.4** | **95.6** | **1.0** | **85.4** | **93.8** | **95.8** | **1.0** |

Table 3: Different methods on CSL-Daily [65] dataset.

### 4.2 Comparison with State-of-the-Art Methods

We compare our framework SEDS on various datasets, including How2Sign, PHOENIX-2014T, and CSL-Daily, with previous works *i.e.* SPOT-ALIGN [15] and CiCo [10], where SPOT-ALIGN builds the final combination (COMB) model by integrating its primary cross-modal (CM) model and sign recognition (SR) model.

For the How2Sign dataset, Table 1 shows the performance comparison between SEDS and existing methods. Benefiting from the large-scale image-text pre-training model CLIP [50], CiCo and our method SEDS achieve significant performance improvements compared to SPOT-ALIGN. These consistent performance improvements demonstrate that utilizing prior visual knowledge of CLIP and contrastive learning paradigms greatly enhances the ability of model in visual-language alignment. Compared to current strongest competitor CiCo, SEDS achieves 62.5(+6.3) R@1 on sign text-to-video retrieval task, and 57.9(+6.3) R@1 on sign video-to-text retrieval task. The outstanding performance demonstrates the effectiveness of our SEDS, which pays attention to both global visual information and local action information, and enables the most representative features of two modalities to fully interact and fuse.

In Tables 2 and 3, we evaluate the performance of SEDS on the PHOENIX-2014T and CSL-Daily datasets. Our model achieves a significant improvement of 76.8(+7.3), 85.8(+10.5) T2V R@1, and 78.7(+8.5), 85.4(+10.7) V2T R@1 on the PHOENIX-2014T and CSL-Daily datasets, respectively. These results indicate that SEDS has good generality and robustness in various sign language languages from different countries and regions.

### 4.3 Ablation Studies

**Performance of Sign Encoders with Different Modalities.** In Table 4, we report the performance of Pose encoder and RGB encoder individually on How2Sign, PHOENIX-2014T, and CSL-Daily datasets. We observe that Pose encoder performs better on CSL-Daily dataset, while RGB encoder has higher performance on PHOENIX-2014T dataset. As a professionally recorded dataset of daily life sign language, CSL-Daily has clearer and more standardized human body details and movements, and the differences between different sign gestures are significant, so it is more necessary to pay attention to local hand movements. PHOENIX-2014T

| Structure Pose | RGB | Dataset | T2V | | | V2T | | |
|---|---|---|---|---|---|---|---|---|
| | | | R@1↑ | R@5↑ | R@10↑ | R@1↑ | R@5↑ | R@10↑ |
| ✓ | | How2Sign [16] | 55.9 | 69.6 | 74.9 | 50.9 | 64.7 | 69.3 |
| ✓ | | PHOENIX-2014T [3] | 65.0 | 86.4 | 92.1 | 65.3 | 86.0 | 92.2 |
| ✓ | | CSL-Daily [65] | 80.5 | 90.9 | 94.0 | 80.0 | 89.5 | 92.9 |
| | ✓ | How2Sign [16] | 54.3 | 68.8 | 74.4 | 48.3 | 62.6 | 68.7 |
| | ✓ | PHOENIX-2014T [3] | 70.4 | 88.6 | 94.4 | 70.1 | 88.8 | 94.5 |
| | ✓ | CSL-Daily [65] | 75.4 | 88.3 | 92.4 | 73.5 | 87.7 | 92.3 |

**Table 4: The ablation study on How2Sign to investigate the influence of encoder architecture.**

| Structure | T2V | | | V2T | | |
|---|---|---|---|---|---|---|
| | R@1↑ | R@5↑ | R@10↑ | R@1↑ | R@5↑ | R@10↑ |
| Add-MLP | 60.4 | 73.5 | 78.5 | 55.5 | 68.9 | 73.6 |
| Concate-MLP | 60.0 | 72.9 | 76.7 | 55.2 | 67.6 | 72.8 |
| Concate-Trans | 58.1 | 71.5 | 75.9 | 53.5 | 66.7 | 71.6 |
| Cross-Atten | 58.9 | 72.4 | 76.4 | 54.3 | 67.2 | 72.3 |
| **CGAF** | **62.5** | **75.1** | **80.1** | **57.9** | **70.4** | **74.9** |

**Table 5: The ablation study on How2Sign to investigate the influence of different fusion modules.**

as a sign dataset collected from German weather forecasts TV program, focuses more on the specific field of weather. So its sign gestures are more similar and need to be combined with global visual signals such as facial expressions for further semantic understanding. The content of these two datasets exactly corresponds to the conditions where Pose and RGB modalities are applicable. For How2Sign sign language dataset, the performance of Pose encoder and RGB encoder is basically similar. As the largest sign language dataset with a wide variety of topics, How2Sign needs to emphasize both local action details and global visual signals. This is also in line with the design intention of our framework SEDS, which has demonstrated outstanding performance on all three datasets.

**Different Feature Fusion Methods.** To validate the effectiveness of our introduced Cross Gloss Attention Fusion (CGAF) Module in the fusion of Pose and RGB features, we compare it with several other fusion methods in Table 5. Add-MLP adds the sequence features of two modalities at corresponding positions and passes them to a Multi-Layer Perception (MLP), resulting in the second-best performance. Concate-MLP method replaces the *add* operation with *concatenate*, and its performance is slightly inferior compared to the Add-MLP. Concate-Trans concatenates the two sequence features and feeds them into a shallow-layer Transformer for interaction, then separates them for addition at corresponding positions. This method exhibits a notable decrease in performance when compared to the first two methods, as the feature aggregation at specific positions incorporates irrelevant distant features from both modalities. Similarly, Cross-Atten also aggregates irrelevant distant features from the opposite modality, leading to a certain decline in performance. The CGAF Module effectively integrates only local features with similar semantics surrounding corresponding positions from inter-modal and intra-modal while ignoring distant features. As a result, it achieves the best performance among various fusion methods.

**Analysis of Pose-RGB Fine-grained Matching Objective.** In our framework, the role of the Pose-RGB fine-grained matching

| Pose-RGB Alignment Yes | No | T2V | | | V2T | | |
|---|---|---|---|---|---|---|---|
| | | R@1↑ | R@5↑ | R@10↑ | R@1↑ | R@5↑ | R@10↑ |
| | ✓ | 60.8 | 74.7 | 79.7 | 55.8 | 68.7 | 74.2 |
| ✓ | | **62.5** | **75.1** | **80.1** | **57.9** | **70.4** | **74.9** |

**Table 6: The ablation study on How2Sign to investigate the influence of Pose-RGB alignment on CGAF.**

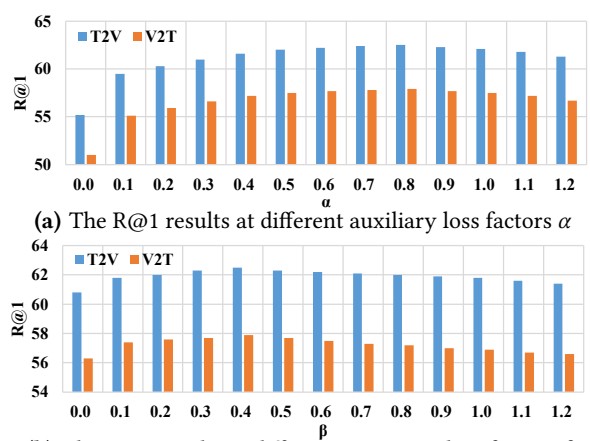

**(a)** The R@1 results at different auxiliary loss factors $\alpha$

**(b)** The R@1 results at different Pose-RGB loss factors $\beta$

**Figure 5: The ablation study on How2Sign to investigate the influence of $\alpha$ and $\beta$. Where $\alpha$=0.8 in (a) and $\beta$=0.4 in (b).**

objective is to explicitly align the corresponding Pose and RGB clip features before fusion, then implicitly reduce the difference between distribution of Pose-Text and RGB-Text fine-grained matching matrices. We demonstrate the effect of this objective on our CGAF module in Table 6. CGAF module benefits significantly from Pose-RGB fine-grained alignment. This objective not only enhances the fusion of corresponding position features, but also leads to better aggregation of semantically similar neighbors from two modalities in the latent space. Consequently, the performance improvement of the CGAF method is +1.7 T2V and +2.1 V2T R@1.

**The Effect of Factors $\alpha$ and Factors $\beta$.** As depicted in Fig. 5, we investigate the effect of the auxiliary loss factor $\alpha$ and the Pose-RGB matching factor $\beta$ on T2V and V2T tasks. For the auxiliary loss factor $\alpha$, there is a significant improvement in performance when it increases from 0 to 0.1. The performance continues to rise steadily with the increase of $\alpha$, reaching a peak at 0.8, after which it gradually decreases. This indicates that the existence of auxiliary loss effectively provides supervision for both branches. However, an overly large auxiliary loss may overly emphasize the training of two branches. For the Pose-RGB matching factor $\beta$, performance consistently improves until it reaches 0.4, after which it begins to decline gradually. A smaller value enables fine-grained alignment between Pose and RGB clip features, while an excessively large $\beta$ will overemphasize this objective, compromising the representation ability of Pose and RGB modalities.

## 4.4 Qualitative Results

**Visualization of the Fine-grained Similarity Matrices.** To provide a more intuitive demonstration of the effect of Pose-RGB fine-grained matching objective, we select a pair of sign language video

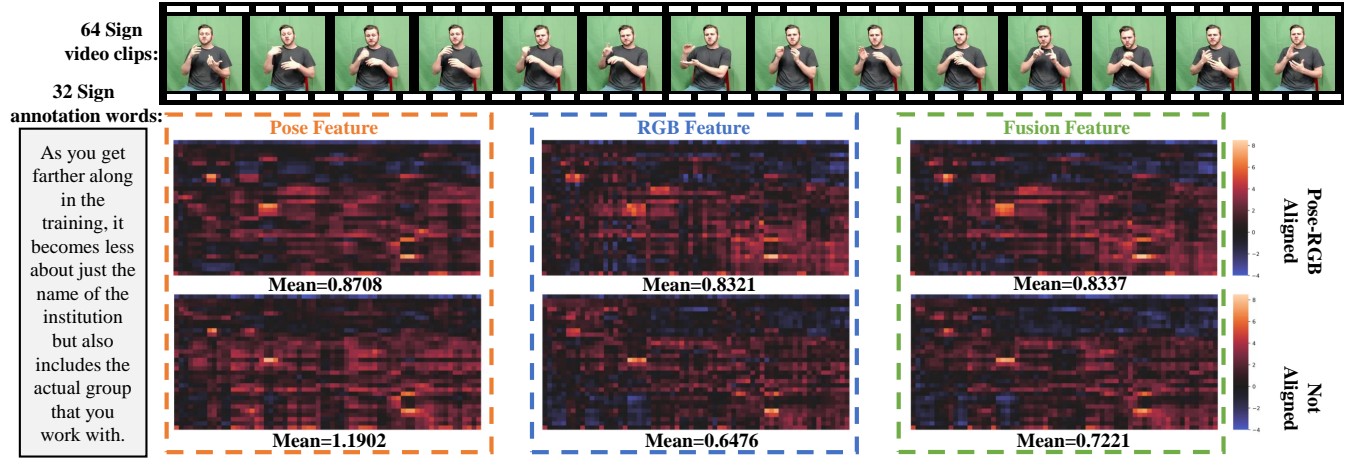

**Figure 6: Visualization of the fine-grained similarity matrices of 64 clips in the video and 32 words in the text.**

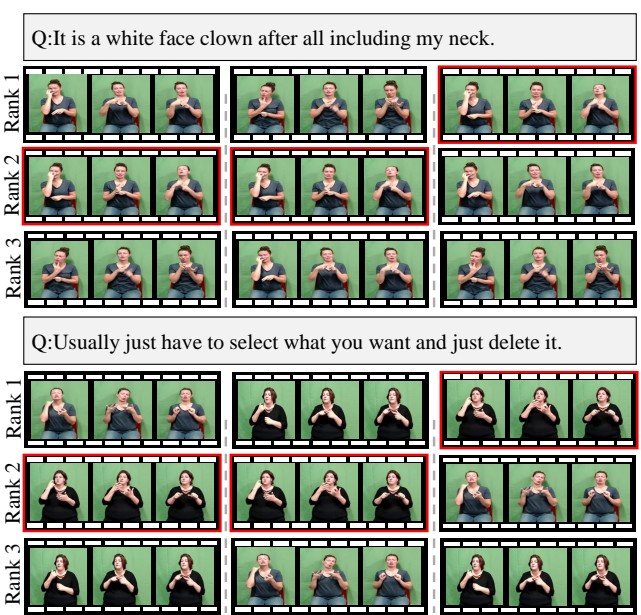

**Figure 7: Top-3 TVR results on How2Sign. Left: by Pose encoder. Middle: by RGB encoder. Right: by our framework. Red: correct videos. Q: query annotation.**

and annotation. Then we calculate and display the fine-grained similarity matrices between the Pose, RGB, and Fusion features with text in the form of heat maps, both with and without this objective. As shown in Fig. 6, the heat maps in the upper part of the figure represent the matrices when the Pose-RGB matching objective exists, while those in the lower part represent the situation when the objective is absent. The distribution differences between the Pose and RGB similarity matrices are smaller in the upper part than in the lower part. Additionally, the mean values of the distributions are essentially the same for the upper matrices, while they differ significantly for the lower matrices. The Fusion similarity matrix in the upper part contains more high similarity scores compared to the lower part, and the overall distribution has a higher mean

value. This observation suggests that the presence of the Pose-RGB matching objective allows for the fusion of features to consider the characteristics of both the Pose and RGB features, enabling them to better match the corresponding words in sign language annotation. In contrast, the focus during fusion tends to favor a single modality with higher mean scores of fine-grained matrix when the objective is absent, resulting in poor fusion performance.

**Retrieval Results of Different Sign Encoders.** The Top-3 TVR results are displayed in Fig. 7. Specifically, in the first example, we observe that the Pose encoder focuses more on the similarity of detailed hand movements and key joint positions, while the RGB encoder emphasizes the overall action and expression of the person. In the second example, which includes two different signers, the RGB encoder shows a clear preference for matching specific individuals, even though there are significant differences in the details of local part movements. The Pose encoder, due to its input modality, eliminates this potential bias. By combining the strengths of both the Pose modality and the RGB modality, our framework SEDS retrieves the correct results in both examples. It is capable of focusing on both local details and the global semantics of the videos, while also exhibiting great robustness and generalization.

## 5 CONCLUSION

In this paper, we propose a novel framework called Semantically Enhanced Dual-Stream Encoder (SEDS) for the emerging Sign Language Retrieval task. SEDS uses online Pose encoder and offline RGB encoder to extract features from sign language videos, focusing on local action details and global visual semantics simultaneously. In the fusion stage, we propose a Cross Gloss Attention Fusion (CGAF) Module to fuse local information with similar semantic information from two modalities. We also develop a supervised Pose-RGB fine-grained matching objective, to match the contextual fine-grained dual-stream features. Our SEDS outperforms previous work significantly on How2Sign, PHOENIX-2014T, and CSL-Daily datasets. We hope that our work helps future exploration of sign language video retrieval tasks and promotes the development of multimodal alignment in retrieval tasks.

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
