# OpenReview forum: "SEDS: Semantically Enhanced Dual-Stream Encoder for Sign Language Retrieval"
_acmmm.org/ACMMM/2024/Conference — MM2024 Poster_

### Official Review · Reviewer_ihLz · 2024-05-18

**Rating:** 4
**Confidence:** 3

**Summary:**

This paper proposes a sign language retrieval framework by utilizing extra human pose signal. In detail, authors design a cross gloss attention fusion (CGAF) module to aggreggate video clip features and human pose features. Besides, they use a Pose-RGB fine-grained matching objective to align two visiual modalities. Extensive experiments are conducted to show the effectiveness and efficiency of the proposed framework.

**Strengths:**

1. The paper is well-written and easy to follow.
2. The idea to introduce human pose signal is very reasonable for sign language retrieval.
3. The performance improvement of the method presented in this paper is significant.
3. The authors have conducted a lot of experiments to prove the validity and rationality of design modules in detail.

**Limitations:**

1. Compared with other methods, the proposed framework uses extra human pose signal. It is better to provide experimental results of other methods equipped with human pose signal. In other words, I wonder if the introduction of human pose signal could be used as a general method to improve the performance of sign language retrieval models.

**Suitability:**

3

---

### Official Review · Reviewer_rCop · 2024-05-20

**Rating:** 3
**Confidence:** 4

**Summary:**

This paper proposes a framework to effectively reduce the huge memory cost of dense visual data embedding in end-to-end training, proposing a Cross Gloss Attention Fusion (CGAF) module and effectively enhancing the representation ability of fusion clip features.

**Strengths:**

introduce the skeleton modality

**Limitations:**

1. The method section involves an excessive number of variables.
2. Visualize the two jointly optimized losses mentioned in the Fig.2.
3. The experimental section currently employs only two comparison methods. Could you provide a rationale for this choice? Furthermore, incorporating additional control experiments would strengthen the validity and reliability of the findings.
4.I wonder the motivation of introducing the skeleton modality, how to choose the skeleton detectors? What are the different influences with RTMPose openpose and mmpose?
5.The organization of the methods section is confusing to me. The specific description of the proposed method is limited, whereas a significant amount of space is devoted to existing technologies such as infoNCE.

**Suitability:**

2

---

### Official Review · Reviewer_eaDD · 2024-06-01

**Rating:** 3
**Confidence:** 3

**Summary:**

The paper presents a novel framework for sign language retrieval called the Semantically Enhanced Dual-Stream Encoder (SEDS). It integrates Pose and RGB modalities to capture both local and global information from sign language videos. The proposed framework includes a Cross Gloss Attention Fusion (CGAF) module to aggregate semantically similar features from both modalities and a Pose-RGB Fine-grained Matching Objective to enhance the representation of the fused features.

**Strengths:**

1. Innovative Multi-Modal Approach: The paper introduces the Semantically Enhanced Dual-Stream Encoder (SEDS) framework, which uniquely integrates Pose and RGB modalities to capture both local action details and global semantic information, significantly enhancing the feature representation for sign language videos.

2. Effective Feature Fusion: The Cross Gloss Attention Fusion (CGAF) module and Pose-RGB Fine-grained Matching Objective are novel contributions that improve the fusion of semantically similar features and the contextual alignment of multi-modal features, leading to superior performance in sign language video retrieval tasks.

3. Comprehensive Experimental Validation: Extensive experiments on multiple datasets (How2Sign, PHOENIX-2014T, CSL-Daily) demonstrate that the proposed method significantly outperforms state-of-the-art approaches, validating its effectiveness and generalizability across different sign languages and scenarios.

**Limitations:**

1. Lack of Motivation for Multi-Modal Data: While the introduction of Pose data aims to enhance feature representation, the paper does not sufficiently justify the necessity of multi-modal data over single-modality approaches. A more detailed explanation of why existing RGB-only methods are inadequate and how Pose data specifically addresses these shortcomings would strengthen the motivation.

2. Fairness in Experimental Comparisons: The introduction of Pose data introduces a new modality, raising concerns about the fairness of comparisons with existing methods that rely solely on RGB data. The paper should ensure that baseline methods are adapted to incorporate similar types of information or provide a detailed discussion on the limitations of single-modality approaches in capturing the required features.

3. Resource Consumption Analysis: The paper does not discuss the potential increase in computational resource consumption due to the additional data stream. An analysis of GPU memory usage, processing time, and computational complexity is necessary to provide a comprehensive evaluation of the method’s efficiency.

**Suitability:**

2

---

### Official Review · Reviewer_MTjh · 2024-06-03

**Rating:** 4
**Confidence:** 3

**Summary:**

This paper introduces a new sign language retrieval framework called Semantically Enhanced Dual Stream Encoder (SEDS). The SEDS framework utilizes online pose encoders and offline RGB encoders to extract features from sign language videos, while paying attention to local action details and global visual semantics. In the fusion stage, a Cross Gloss Attention Fusion (CGAF) module is proposed to fuse local information with similar semantic information between two modalities. In addition, a supervised Pose RGB fine-grained matching target has been developed for matching fine-grained dual stream features in the context. The experimental results of SEDS on multiple datasets such as How2Sign, PHOENIX-2014T, and CSL-Daily, are significantly better than previous works. The author hopes that this work can promote the exploration of future sign language video retrieval tasks and the development of multimodal alignment in retrieval tasks.

**Strengths:**

1. The structure of this paper is clear and easy to follow.
2. In response to the existing problems in sign language retrieval work, the SEDS framework effectively solves the problem of local action details being overwhelmed by a large amount of visual information redundancy by introducing posture modal knowledge.
3. Through the Cross Gloss Attention Fusion (CGAF) module, effective aggregation of similar semantic information between two video modalities has been achieved, which helps to enhance the semantic representation of visual words in sign language video editing.
4. Extensive experiments on the How2Sign, PHOENIX-2014T, and CSL-Daily datasets have shown that the SEDS framework is significantly superior to existing state-of-the-art methods

**Limitations:**

1. Although the framework includes lightweight networks, pose estimation and dual stream encoders may require higher computational resources, and the article lacks explanation of the possible increase in training or inference time.
2. There is no explanation in the paper whether it will release source code.

**Suitability:**

3

---

### Meta-Review · Area_Chair_VUAY · 2024-06-30

**Recommendation:** Accept (Poster)
**Confidence:** 4

**Metareview:**

The authors provide some new arguments to address the critical issues highlighted by reviewers. Most of the reviewers are satisfied with the response. I recommend that the authors incorporate the reviewers' insightful feedback into their revision.

---

### Meta-Review · Senior_Area_Chairs · 2024-07-10

**Recommendation:** Accept (Poster)
**Confidence:** 4

**Metareview:**

This paper received mixed ratings initially. After rebuttal, three reviewers tend to accept the paper while one who gave BR still have some concerns. SAC and AC checked the paper,reviews and rebuttal and recommend acceptance of the paper.